# Preparation and In Vitro/In Vivo Evaluation of Orally Disintegrating/Modified-Release Praziquantel Tablets

**DOI:** 10.3390/pharmaceutics13101567

**Published:** 2021-09-27

**Authors:** Xuemei Wen, Zhaoyou Deng, Yangfeng Xu, Guoqing Yan, Xin Deng, Liqin Wu, Qiuling Liang, Fang Fang, Xin Feng, Meiling Yu, Jiakang He

**Affiliations:** 1College of Animal Science and Technology, Guangxi University, Nanning 530004, China; xuemeiwen1992@163.com (X.W.); pharmacologyvip@163.com (Z.D.); yangfengxu1992@gmail.com (Y.X.); ygq18176261001@163.com (G.Y.); 1718393003@st.gxu.edu.cn (X.D.); liqinwu16@126.com (L.W.); qlliang1210@163.com (Q.L.); ffang@gxu.edu.cn (F.F.); 2Department of Pharmaceutics and Drug Delivery, School of Pharmacy, The University of Mississippi, Oxford, MS 38677, USA; xinfeng2017@gmail.com

**Keywords:** praziquantel, hot-melt extrusion, orally integrating tablets, modified-release, pharmacokinetics

## Abstract

This study was designed to develop orally disintegrating/sustained-release praziquantel (PZQ) tablets using the hot-melt extrusion (HME) technique and direct compression, and subsequently evaluate their release in in vitro and in vivo pharmacokinetics. For the extrusion process, hypromellose acetate succinate (HPMCAS)-LG was the carrier of pure PZQ, with a standard screw configuration used at an extrusion temperature of 140 °C and a screw rotation speed of 100 rpm. Differential scanning calorimetry (DSC), thermogravimetric analysis (TGA), powder X-ray diffraction (PXRD) and Fourier-transform infrared spectroscopy (FTIR) were performed to characterize the extrudate. Orally disintegrating/sustained-release praziquantel tablets (PZQ ODSRTs) were prepared by direct compression after appropriate excipients were blended with the extrudate. The release amount was 5.10% in pH 1.0 hydrochloric acid at 2 h and over 90% in phosphoric acid buffer at 45 min, indicating the enteric-coating character of PZQ ODSRTs. Compared with the pharmacokinetics of marketed PZQ tablets (Aipuruike^®^) in dogs, the times to peak (T_max_), elimination half-life (t_1/2λ_) and mean residence time (MRT) were extended in PZQ ODSRTs, and the relative bioavailability of PZQ ODSRTs was up to 184.48% of that of Aipuruike^®^. This study suggested that PZQ ODSRTs may have potential for the clinical treatment of parasitosis.

## 1. Introduction

Parasitic diseases, especially schistosomiasis, cysticercosis and echinococcosis, are common diseases in dogs and cats, some of which are seriously harmful to animal growth and health and even play a crucial role in transmission between humans and domestic livestock [1]. Schistosomiasis is distributed in 76 countries and regions [2], such as sub-Saharan Africa, parts of the Congo River and some developing countries in Asia and Brazil [3], where thousands of people have experienced impacts on their health and wellbeing because of helminth infections with blood flukes [4]. Echinococcosis is commonly found in dogs, and is the definitive host [5]. Cerebral cysticercosis and multiheaded mites parasitizing the host brain could cause neurological symptoms. This parasite may bring potential danger to the development of the breeding industry, and it could damage the health of people by parasitizing meat products or pets of human beings [6]. For the above-mentioned diseases, maintaining a clean and comfortable dwelling environment and expelling parasites via drug treatment is a preferable way to control the prevalence of parasitosis in animals.

Praziquantel was listed by the World Health Organization as an essential medicine for treating schistosomiasis, clonorchiasis, opisthorchiasis, tapeworm infections, cysticercosis and hydatid disease [7]. Specifically, PZQ could become the first choice for the treatment of various parasitic diseases due to its advantages of low cost, high efficacy [8], low dosage [9], short treatment course [10], fast metabolism and low toxicity. In the 1970s, PZQ was developed as a new drug for the clinical treatment of parasitic diseases [11]. PZQ formulations currently on the market for dogs include injections [12], tablets [13] and transdermal preparations [14]. Owing to the low water solubility and strong permeability [15], PZQ injections have the disadvantage of a low effective concentration, a large injection volume and poor stability. In comparison to injection, oral dosage forms are more popular because of their easy administration, transportation and storage. Nevertheless, some oral dosage forms of PZQ are not only difficult to control in terms of dosage and accurate administration (such as with powders), but also have a strong first-pass effect and low bioavailability [16], which weakens the therapeutic effect of PZQ. Moreover, it is difficult for dogs suffering from dysphagia to swallow an intact tablet. Consequently, research and development of novel dosage forms of PZQ and improved bioavailability and clinical efficacy are of great significance for the prevention and treatment of zoonotic parasitic disease. To overcome the above challenges, orally disintegrating tablets (ODTs) with modified-release PZQ properties were designed and developed in this study.

ODTs are novel dosage forms that combine the benefits of both conventional tablets and liquid formulations. ODTs disintegrate rapidly without the aid of water when placed in the oral cavity [17], thus improving the compliance of dogs with dysphagia. ODTs have been found to be the preferred dosage form for patients with nausea [18], vomiting [19] or motion sickness [20]. Different techniques, such as conventional tablet pressing [21], freeze-drying [22], hot-melt extrusion (HME) [23] and melt-adsorption [24], are employed for manufacturing ODTs, and HME was utilized in this study. HME has been applied to a wide variety of pharmaceutical dosage forms, including granules [25], pellets [26], films [27], tablets [28] and implants [29]. It is widely accepted that HME is easy to scale up and operate continuously as a solvent-free and energy-saving approach for the preparation of various pharmaceutical systems [30]; moreover, HME can not only enhance the solubility and bioavailability of poorly soluble active pharmaceutical ingredients (APIs), it also improves their stability and bitter taste [31] and is especially suitable for preparing ODTs.

To the best of our knowledge, there have been no marketed PZQ ODTs prepared via HME technology. Therefore, in this study, HME was utilized to prepare solid dispersions of PZQ that were then mixed with suitable excipients for direct compression into sustained-release ODTs. The release rates in in vitro and in vivo pharmacokinetic studies of ODTs were evaluated to provide a basis for further development of formulation applications for dogs.

## 2. Materials and methods

### 2.1. Materials

A PZQ standard was purchased from the China Institute of Veterinary Drug Control (Beijing, China). Pure PZQ (98.5%) was ordered from Beijing Ouhe Technology Co., Ltd. (Beijing, China). AquaSolveTM hypromellose acetate succinate (HPMCAS)-LG was purchased from Ashland Specialty Ingredients (Wilmington, DE, USA). Microcrystalline cellulose, mannitol, low-substituted hydroxypropyl cellulose and magnesium stearate were ordered from Shanghai Yuanye Bio-Technology Co., Ltd. (Shanghai, China). High-performance liquid chromatography (HPLC)-grade acetonitrile, methanol, dichloromethane and methyl-tert-butyl ether were obtained from Fisher Scientific (St. Louis, MO, USA). Marketed Aipuruike^®^ PZQ tablets (Aipuruike^®^) were purchased from Beijing Agrochemical Pharmaceutical Co., Ltd. (Beijing, China). All other chemical reagents used for HPLC, dissolution and pharmacokinetics were analytically pure.

### 2.2. Extrudate Preparation

PZQ (20% *w*/*w*) was mixed with HPMCAS utilizing a V-shell blender (GlobePharma, Maxiblend^®^, New Brunswick, NJ, USA) after sieving through a size #30 USP mesh. The blended materials were extruded on a 16 mm co-rotating, twin-screw extruder (Prism Euro Lab., Thermo Fisher Scientific, Waltham, MA, USA) with a screw rotating speed of 100 rpm and an extrusion temperature of 140 °C to manufacture PZQ extrudates. The extrudate was cooled to room temperature and further comminuted in a pulverizer and then sieved to suitable particle sizes ranging from 350 to 500 μm (#35 to #60). The milled and sieved extrudates were stored in brown glass vials with foil-lined caps for future use.

### 2.3. Thermogravimetric Analysis (TGA)

TGA was used to determine the thermal stability of the API and select the processing temperatures of the carriers (TGA, NETZSCH STA 449 F3 Jupiter^®^, Selb, Bavaria, Germany). The data were analyzed using Proteus software. Pure PZQ was evaluated for thermal stability at high temperatures. Approximately 3–5 mg of the sample was weighed and heated from 30 to 300 °C under an inert nitrogen atmosphere at a heating rate of 20 °C/min for thermogravimetric analysis. The TGA information of HPMCAS was provided by the supplier.

### 2.4. Differential Scanning Calorimetry (DSC) Analysis

Thermal analysis was utilized to determine the physical state of the API in the physical mixture and extrudate using a DSC-404F3 instrument (NETESCH, Selb, Bavaria, Germany) in an inert nitrogen atmosphere. The physical states of pure PZQ, HPMCAS, the physical mixture and the extrudate were studied. An exactly weighed amount (3–5 mg) of each sample was put in a hermetically sealed aluminum pan and heated from 30 to 200 °C at a rate of 20 °C/min. The Pyris application (Shelton, CT, USA) was applied for data processing and analysis.

### 2.5. Powder X-Ray Diffraction (PXRD)

PXRD measurements were used to study the crystallinity of PZQ in the hot-melt extrudate. Under normal conditions, a PXRD apparatus (RIGAKU, Tokyo, Japan) was employed to observe the crystal forms of the API with CuKα radiation at 15 mA and 30 kV, with a scan rate of 4°/min and diffraction angle (2θ) range of 1–40°.

### 2.6. Fourier-Transform Infrared (FTIR) Spectroscopy

The FTIR spectra of the API were recorded by a Nicolet 6700 FTIR spectrometer (Thermo Fisher Scientific) in the range of 400–4400 cm^−1^ to determine any potential interactions between PZQ and the carrier in the physical mixture or extrudate.

### 2.7. Drug Content Analysis

A Waters HPLC system was used to determine the drug content. The system configuration included an e2695 separations module, a Waters 2998 photodiode array detector and a Waters 717 plus autosampler (Waters Technologies Corporation, 34 Maple St., Milford, MA 0157, USA). The chromatographic conditions were as follows: detection wavelength, 220 nm; mobile phase, acetonitrile/Milli-Q water (60/40, *v*/*v*); flow rate, 1.0 mL/min; injection volume, 20 μL; column, reversed-phase Inertsil ODS-3 C18 (250 × 4.6 mm; inside diameter, 5 μm) (GL Sciences Inc., Tokyo, Japan). HPLC data were processed with Empower V. Software (Milford, MA, USA), and all analyses were performed in triplicate (*n* = 3).

### 2.8. Preparation of PZQ ODSRTs

PZQ ODSRTs were prepared by the direct compression method using an infrared tablet press PC-12J (Jingtuo Instrument Technology Co., Ltd., Tianjin, China). The optimum formulas are displayed in Table 1. After homogeneously mixing the extrudate with the tableting excipients, the mixture was poured into the feed port and semiautomatically pressed into tablets. Each tablet weighed 1000 ± 50 mg and contained 50 ± 2.5 mg of PZQ.

### 2.9. In Vitro Drug Release

A dissolution test was applied to measure the simulated in vivo oral release of pure PZQ and PZQ ODSRTs on the Hanson SR8-plus™ dissolution apparatus (Chatsworth, Los Angeles, CA, USA). The paddle rotation speed was 100 rpm, and the dissolution vessel was filled with 250 mL of dissolution medium.

The tablets were put in 0.1 N hydrochloric acid solution maintained at 37 ± 0.5 °C (*n* = 6) and pH 6.8 phosphate buffer solution (*n* = 6) maintained at 37 ± 0.5 °C to detect their dissolvability in different media. Each sample containing 50 mg of PZQ was poured into the dissolution vessel. A sample solution aliquot (3 mL) was extracted at each time point, and the same volume of fresh medium was placed back into the dissolution vessel. The extraction solution was pretreated to detect the concentration of PZQ in 0.1 N HCl and pH 6.8 phosphate buffer solutions, and all samples were analyzed by a Waters HPLC system.

### 2.10. Disintegration Time

The disintegration time of the PZQ ODSRTs was measured by a disintegration time measuring instrument (Tianda Tianfa, Tianjin, China). A basket was immersed in a 750 mL beaker and located 25 mm from the bottom of the beaker. The temperature of the water in the beaker was 37 ± 0.5 °C, and the height of the water level was adjusted. When the basket was raised, the screen was 15 mm below the water surface. The disintegration time required for the tablet to completely disintegrate into fine particles was noted. Measurements were carried out in six tablet replicates (*n* = 6), and the mean ± SD was recorded.

### 2.11. Friability and Hardness Test

The tablet friability was tested using a friability tester (Tianda Tianfa, Tianjin, China). According to the USP, when the weight of the tested tablets exceeded 0.65 g, 10 tablets were revolved together 100 times in the rotating wheel. The tablets should be carefully dedusted and accurately weighted prior to testing. After the test, the tablets were reweighed, and the weight loss was calculated (Equation (1)). The percentage weight loss should not exceed 1% to consider the batch qualified.
Weight loss % = (initial weight − weight after test)/(initial weight) × 10 (1)

The hardness of the six tablets was assessed by a hardness tester (Tianda Tianfa, Tianjin, China) to examine the mechanical strength of the prepared tablets.

### 2.12. Stability Studies

The accelerated stability and long-term stability of the prepared tablets were determined to evaluate the stability of the formulation in terms of physical properties, disintegration time and chemical content. For accelerated stability, samples were deposited in a stability chamber at 40 ± 2 °C and a relative humidity (RH) of 75% ± 5% and withdrawn to estimate the physical appearance, determine the disintegration time and analyze the content of API via HPLC after 0, 1, 2, 3 and 6 months. Samples were placed under long-term storage conditions at 30 °C and an RH of 65% with predetermined intervals (0, 3 and 6 months) to obtain information on the long-term stability. The evaluation method was similar to the accelerated stability.

### 2.13. Pharmacokinetic Analysis

Six beagles, three males and three females, weighing approximately 10 kg, were obtained from Beijing Rixin Technology Co., Ltd. (Beijing, China). All dogs were clinically examined to ensure that they were in good health for this experiment. The beagles had not been previously administered any antiparasitic drugs to treat or prevent parasitic infection. The animals were in optimum health and had free access to food but were fasted for 12 h before PZQ administration. Aipuruike^®^ and PZQ ODSRTs were orally administered at a dose of 5 mg of PZQ/kg per dog. Blood (2 mL) was drawn at different time points (0, 0.083, 0.25, 0.5, 1, 2, 4, 6, 8, 12 and 24 h post-dosing), temporarily stored in heparinized centrifugal tubes and centrifuged at 3000 rpm for 10 min to separate the plasma. Plasma free of PZQ was collected and used as a blank. The blank plasma samples were stored at −20 °C until further analysis to ensure the elimination of PZQ in plasma. All 6 animals were subjected to a randomized 2 × 3 crossover design. The washout period was two weeks.

All procedures involving animals during this experimentation were approved by the Guangxi University Animal Care and Use Committee (protocol number: GXU-2017-021).

#### 2.13.1. Plasma Sample Preparation for Analysis

A 0.5 mL blood sample was homogenized with 1.5 mL of methyl tert-butyl ether/dichloromethane mixture (2:1, *v*/*v*) and centrifuged at 12,000 rpm for 10 min for protein precipitation. The supernatant was transferred to a new centrifuge tube and evaporated to dryness in a water bath at 45 °C with nitrogen. The residue was reconstituted in 0.5 mL of mobile phase, and 50 μL was injected into the HPLC column after filtration with a 0.22 μm organic filter. A reversed-phase column (Luna ODS-3, 250 × 4.6 mm, i.d., 5 μm, Aschaffenburg, Bavaria, Germany) was used for chromatographic separation. The mobile phase consisted of acetonitrile/water with the following gradient program: 0–13 min, 48/52; 13.01–16 min, 48→100/52→0; 16.01–18 min, 100/0; 18.01–20 min, 100→48/0→52; 20–24 min, 48/52. The flow rate was 1.0 mL/min, and the detection wavelength was set at 217 nm.

Linear calibration curves for PZQ in the range of 0.0625–2 μg/mL were generated via a series of blank plasma spiked with six different concentrations. The correlation coefficients (r) were >0.999. Both the inter-day and intraday precision (CV%) were below 10% of the actual value. The mean recovery of PZQ from plasma was 88.2%. The limit of detection (LOD) and limit of quantification (LOQ) were 0.015 and 0.05 μg/mL, respectively.

#### 2.13.2. Statistical Analysis

A noncompartmental analysis was performed by WinNonlin to calculate individual maximum plasma concentration (C_max_), mean residence time (MRT_0–∞_), area under the curve (AUC_0–∞_), elimination half-life (t_1/2λ_) and T_max_ values for each subject. Means ± SD (SD; upper and lower CI 95%) of each parameter were then calculated from individual values.

## 3. Results

### 3.1. Preparation of the Hot-Melt Extrudate

Extrusion processing was performed, with an extrusion temperature of 140 °C, a rotation speed of 100 rpm and drug loading of 20%. The extrudate was smooth and transparent with a glassy yellowish color and possessed an exceptional drug content (>98%) and content uniformity (<3% RSD) after HME processing.

### 3.2. Physical Characterization of the Extrudate

The TGA and DSC results demonstrated that pure PZQ showed no significant degradation at 149.9 °C and exhibited a sharp endothermic peak at 142.5 °C, as shown in Figure 1. The DSC thermograms of pure PZQ, the physical mixture, the PZQ extrudate and HPMCAS are shown in Figure 2. The DSC curves of pure PZQ and the physical mixture exhibited endothermic peaks at approximately 142 °C, which suggested that they had crystalline structures. HPMCAS showed no characteristic peaks, and the endothermic peak for PZQ was not present in the extrudate, indicating that the API became amorphous and uniformly dispersed in the bulk phase of the carrier via the hot-melt extrusion process.

The PXRD peaks of pure PZQ, the physical mixture, the extrudate and HPMCAS are shown in Figure 3. Pure PZQ displayed several characteristic diffraction peaks at 2θ angles between 5° and 40°, indicating that PZQ existed in crystalline form. Even though the intensities were reduced, the same diffraction peaks were still observable in the physical mixture. In comparison with the physical mixture, the diffraction peaks of PZQ were not observed for the extrudate, which suggests that PZQ was converted into a high-energy, amorphous form in the extrudate, and this supported the DSC result, indicating the successful preparation of the amorphous solid dispersion.

The FTIR spectra of pure PZQ, HPMCAS, the physical mixture and extrudate are shown in Figure 4. The results demonstrated that the physical mixture and PZQ hot-melt extrudate had similar characteristic peaks at 2933.1, 1623.7~1646.9 and 1000~1350 cm^−1^, and their FTIR spectra were a simple superposition of the spectra for HPMCAS and pure PZQ. Although the extrudate spectrum exhibited some reductions in peak intensities compared with those of the physical mixture, no new peaks appeared in the extrudate spectrum, indicating that few or no interactions resulted between PZQ and the carrier.

### 3.3. In Vitro Drug Release

As shown in Figure 5 and Figure 6, the amount of PZQ released from the formulation in 45 min exceeded 90% in pH 6.8 phosphate buffer; however, the release rate of the formulation in 0.1 N hydrochloric acid solution was only 5.1%, lower than the 32.01% rate for pure PZQ. More than 90% of the PZQ in Aipuruike^®^ was released in the acidic solution in 2 h, which indicated that the formulation presented enteric-soluble characteristics, but the marketed Aipuruike^®^ did not.

### 3.4. Disintegration Time

According to the provisions of the USA Pharmacopoeia for measuring disintegration times of orally disintegrating tablets, 6 samples should be randomly selected, and the disintegration time should be approximately 30 s or less. The experimental results showed that the time required for disintegration of the formulation from intact tablets to fine particles that could pass through the sieve was 16.84 ± 0.32 s (*n* = 6), well within the 30 s target.

### 3.5. Friability and Hardness Test

The hardness mean was 5.57 ± 0.06 kg, and the friability test demonstrated that the weight loss was 0.85%, which agree with the Chinese Pharmacopeia limits for tablet friability tests.

### 3.6. Stability

Whether in accelerated or long-term conditions, the tablets showed no significant degradation in chemical content, which maintained >95% of intact praziquantel throughout the storage period (Figure 7). All samples had no change in physical properties or appearances.

### 3.7. Pharmacokinetics of the PZQ ODSRTs in Dogs

The mean plasma concentrations and concentration-time profiles (plasma) of different treatment groups after oral administration of 5 mg/kg of PZQ are compared in Figure 8. The relatively low LOQ for the method used in the determination of PZQ plasma concentrations was 50 ng/mL. The pharmacokinetic parameters are provided in Table 2.

In the ODSRTs, the PZQ concentration fell below the detectable limit after nearly 13 h, while that in Aipuruike^®^ was not detectable after 8 h. Compared with those for Aipuruike^®^*,* the elimination half-lives (t1/2λ), T_max_ values and MRT_0–∞_ values of the ODSRTs were prolonged and significantly different (*p* < 0.01), although the maximum plasma concentration (C_max_) of the ODSRTs was not significantly different (*p* < 0.05) from that of Aipuruike^®^. The relative bioavailability of the PZQ ODSRTs was 184.48% ± 54.90%, almost twice that of Aipuruike^®^*,* which demonstrated that the formulation was more effective and exhibited a delayed-release effect.

## 4. Discussion

Presently, the anthelmintic drug PZQ plays a critical role in treating schistosomiasis and other common parasitic diseases. However, as a result of the poor water solubility and first-pass effects of PZQ, commercially available oral formulations are limited because their bioavailabilities require improvement. Xu et al. reported that the oral bioavailability of PZQ in the rice field eel *Monopterus albus* was only 20.9% [32]. Tang et al. compared oral administration of PZQ and its relative bioavailability with that of intramuscular administration in pigs, which indicated that the oral bioavailability of the marketed Aipuruike^®^ was as low as 7.9% [33], which is extremely low. To expand the application of PZQ and improve its pharmaceutical properties, new PZQ formulations with better performances were developed by combining HME and other preparation technologies.

As an innovative and feasible approach, HME [34] has been widely applied in the preparation of various pharmaceutical systems, such as minitablets, granules, immediate- and modified-release tablets and oral fast-dissolving systems. According to the DSC results, the melting point of pure PZQ was 142.5 °C, so the processing temperature in this study was set at 140 °C. At this extrusion temperature, pure PZQ remained stable and did not degrade obviously, as confirmed by TGA. Before HME application, not only should the thermal properties of the API be considered but the glass transition temperature of the carrier should also be close to that of the API. HPMCAS has a glass transition temperature of approximately 120 °C, and the decomposition of HPMCAS begins at approximately 200 °C, which makes it suitable for preparing PZQ extrudates [35]. In addition, HPMCAS not only inhibits the nucleation rates of crystals, but also reduces the growth of crystalloids, which improves storage stability [36,37]. According to its dissolution behavior in aqueous solutions with different pH values, HPMCAS can be divided into three specifications: HG, MG and LG [30]. Comparing three descriptions of HPMCAS, we found that HPMCAS-LG was the polymer with the lowest viscosity and had preferable thermoplasticity. Therefore, we chose HPMCAS-LG as the carrier and 140 °C as the extrusion temperature for preparing the solid dispersion of PZQ.

HME is a heat treatment technology that increases the risk of thermal degradation of the API [38]. Additionally, the extrusion temperature greatly affects the miscibility of the drug and the carrier [39]. DSC, PXRD and FTIR spectroscopy are usually used to evaluate the physical state of the API in excipients. The DSC curve for pure PZQ exhibited a single intrinsic melting point at 142.5 °C, consistent with previous results (141.96 °C) [40]. In the PZQ extrudate, the endothermic peak for PZQ disappeared, indicating that PZQ was in an amorphous or molecular state in the solid dispersion. However, this proposal requires support from PXRD and other techniques. PXRD can be used to qualitatively and quantitatively determine the crystalline form of PZQ in the extrudate. In the pattern of the extrudate, diffraction peaks for PZQ were not observed, which was consistent with DSC results confirming that PZQ encapsulated in the carrier was successfully transformed into an amorphous state. To investigate the molecular interactions between API and its carrier during extrusion processing, FTIR analysis was performed. FTIR spectroscopy can simultaneously show thermally induced changes in components and identify chemical changes in the internal structures of drug molecules. PZQ exhibited characteristic peaks in the FTIR spectrum at 2933.1, 1623.7~1646.9 and 1000~1350 cm^−1^, which were attributed to -CH/-CH_2_/CH_3_, -C=O and -CN groups, respectively. The FTIR results for PZQ were consistent with those described in the literature [41]. It was seen (Figure 4) that the IR spectrum of the extrudate overlapped with those of pure PZQ and HPMCAS and the peak intensities showed obvious decreases, which indicated that there were no interactions between the API and the carrier. The spectrum indicated that amorphous PZQ existed in the extruded solid dispersions prepared by hot-melt extrusion [42].

It is worth noting that in vitro release rates of PZQ from the formulation and from Aipuruike^®^ in different media revealed that Aipuruike^®^ exhibited a relatively high release rate in simulated gastric liquid; however, PZQ in the formulation was slowly released in simulated gastric liquid, and its release rate surpassed 90% in just 45 min in pH 6.8 phosphate buffer solution. This should be attributed to the hot-melt extrusion technology because it converted PZQ and the polymer into an amorphous solid dispersion that prevented PZQ exposure under acidic conditions, which may be conducive to modified-release of the drug to a certain extent [43]. According to the Food and Drug Administration (FDA), ODTs in solid dosage forms contain a medicinal substance or active ingredient that disintegrates quickly within 30 s of placement on the tongue [44], and our disintegration test result was 16.84 ± 0.32 s, which conformed to the standards.

Furthermore, oral absorption of PZQ from the PZQ ODSRTs and Aipuruike^®^ was evaluated in beagles at a dose of 5 mg/kg. Compared with Aipuruike^®^*,* the major pharmacokinetic parameters (t_1/2λ_, MRT_0–∞_, AUC_0–∞_, T_max_, etc.) of the ODSRTs, except for C_max_ (0.43 ± 0.0084 and 0.39 ± 0.28 µg/mL), were significantly different (*p* < 0.01). The T_max_ of the PZQ ODSRTs was delayed, the MRT_0–∞_ was prolonged and the C_max_ of the drug in blood was increased, which indicates that the formulation exhibited better pharmacokinetic behaviors in dogs. The t_1/2λ_ value was 4.28 h in our experiment, which was longer than those of Aipuruike^®^ (1.78 h) and traditional PZQ tablets (generally ranging between 1 and 3 h) [45]. The C_max_ of the formulation was 0.43 ± 0.0084 μg/mL, which was extraordinarily close to the C_max_ (0.49 ± 0.1112 μg/mL) of a novel praziquantel nanocrystal formulation [46]. Furthermore, the relative bioavailability of PZQ ODSRTs was 184.48% ± 54.90%, almost twice that of Aipuruike^®^, suggesting that the formulation was relatively easy to absorb and had a higher absorption degree, which may be related to the better preparation process used for the formulation [47].

In summary, the new PZQ ODSRTs are modified release and enteric coating preparations that could reduce medication frequency, prolong release time and increase bioavailability to provide a better therapeutic effect in the clinical treatment of zoonotic parasite disease.

## 5. Conclusions

In this study, an amorphous solid dispersion of PZQ with HPMCAS was prepared successfully by utilizing HME technology. The amorphous state of the API in the extrudate was confirmed by TGA, DSC, PXRD and FTIR spectroscopy. The extrudate was blended uniformly with excipients and then formulated into orally disintegrating tablets via direct compression. Compared with marketed PZQ tablets, PZQ ODSRTs exhibited rapid disintegration, enteric coating properties and modified release, as well as better pharmacokinetic parameters.

## Figures and Tables

**Figure 1 pharmaceutics-13-01567-f001:**
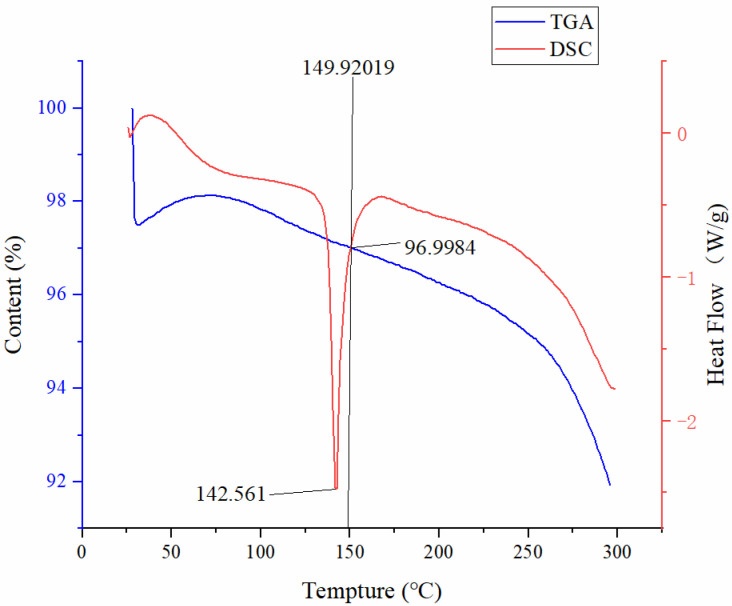
TGA and DSC curves of pure PZQ.

**Figure 2 pharmaceutics-13-01567-f002:**
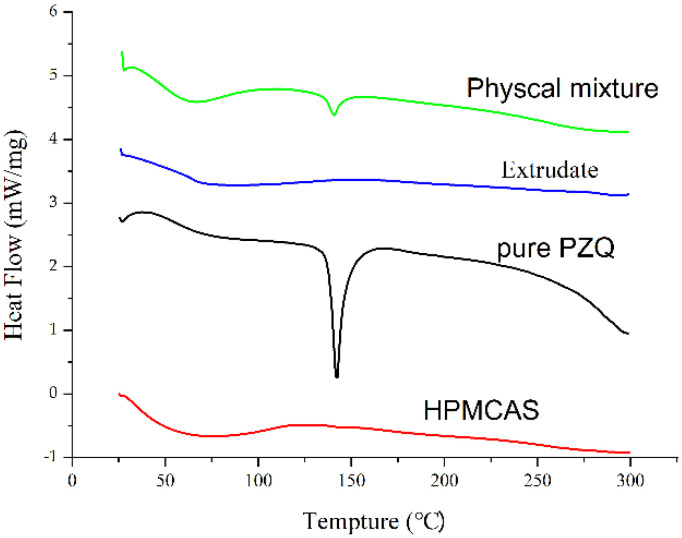
DSC thermograms of pure PZQ, a physical mixture, extrudate and HPMCAS.

**Figure 3 pharmaceutics-13-01567-f003:**
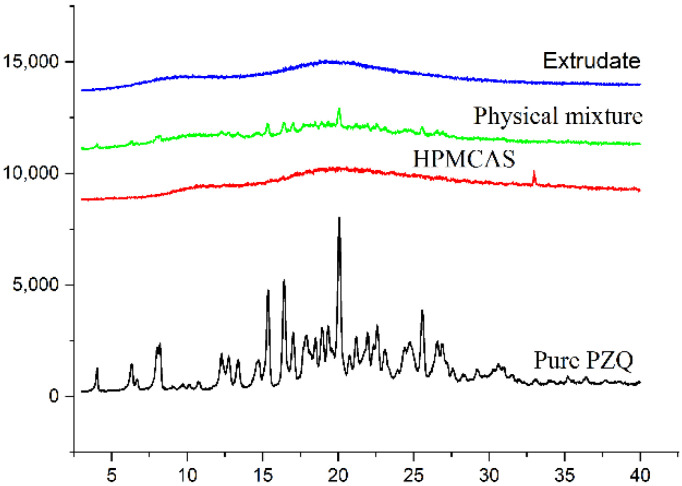
XRD thermograms of pure PZQ, a physical mixture, extrudate and HPMCAS.

**Figure 4 pharmaceutics-13-01567-f004:**
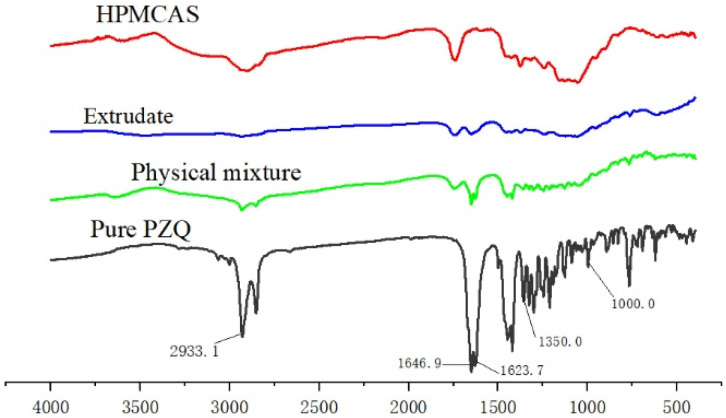
FTIR spectra of pure PZQ, a physical mixture, extrudate and HPMCAS.

**Figure 5 pharmaceutics-13-01567-f005:**
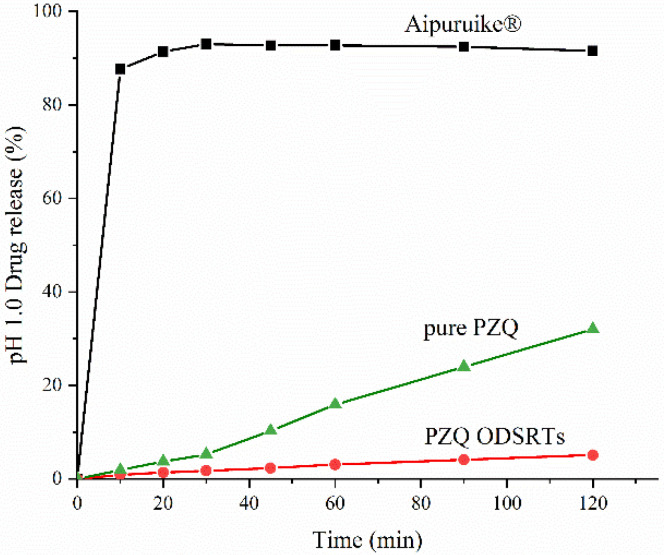
Drug release of PZQ ODSRTs, pure PZQ and Aipuruike^®^ in 0.1 N HCl solution.

**Figure 6 pharmaceutics-13-01567-f006:**
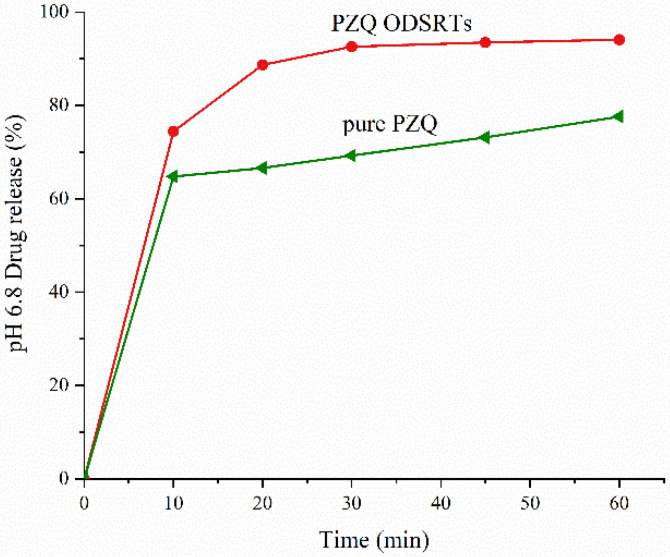
Drug release of PZQ ODSRTs and pure PZQ in buffer medium (pH 6.8).

**Figure 7 pharmaceutics-13-01567-f007:**
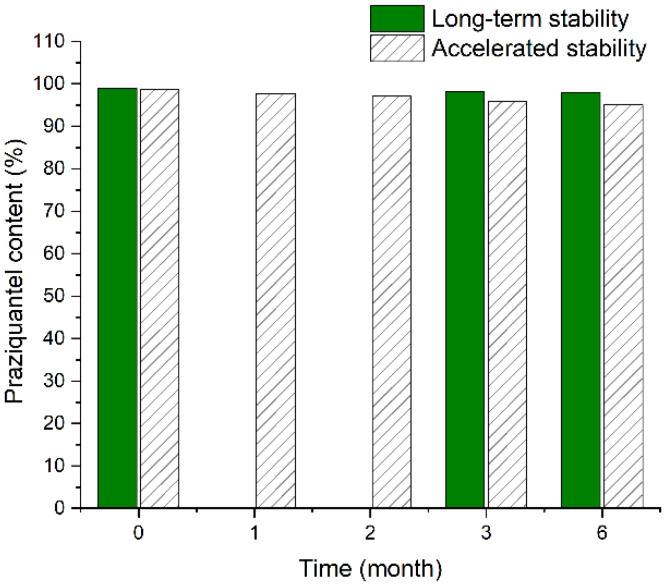
Changes in the chemical content of praziquantel in PZQ ODSRTs under accelerated and long-term conditions.

**Figure 8 pharmaceutics-13-01567-f008:**
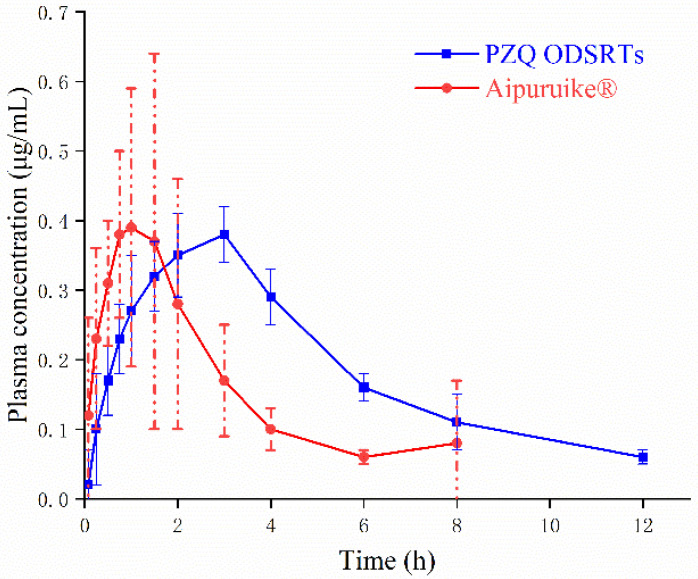
Concentration-time profiles (plasma) of praziquantel following a single p.o. administration of the PZQ ODESRTs and marketed Aipuruike^®^ at a dose of 5 mg/kg body mass (mean ± SD, *n* = 9).

**Table 1 pharmaceutics-13-01567-t001:** Composition of orally disintegrating/sustained-release praziquantel tablets.

Ingredients	Quantity (mg/Tablet)	Function
PZQ hot-melt extrudate	250	active ingredient
Microcrystalline cellulose	187.5	dry binder
mannitol	515	diluent, sweetening agent
Low-Substituted Hydroxypropyl Cellulose	37.5	disintegrant
Magnesium stearate	10	lubricant
Total	1000	

**Table 2 pharmaceutics-13-01567-t002:** Pharmacokinetic parameters of praziquantel following a single p.o. administration of PZQ ODSRTs and Aipuruike^®^ at a dose of 5 mg/kg body mass (mean ± SD, *n* = 9).

	PZQ ODSRTs (p.o.)		Aipuruike^®^		*p*-Value (ANOVA)
t_1/2λ_ (h)	4.28 ± 0.6701		1.78 ± 0.44		0.000
AUC_0–∞_ (μg·h/mL)	2.30 ± 0.28		1.24 ± 0.51		0.000
AUMC_0–∞_ (μg·h^2^/mL)	9.99 ± 1.87		3.13 ± 1.69		0.001
MRT_0–∞_ (h)	4.34 ± 1.221		2.42 ± 0.98		0.001
*C_max_* (μg/mL)	0.43 ± 0.0084		0.39 ± 0.28		0.252
F (%)	184.48 ± 54.90		—		—
*T_max_* (h)	Median	Range	Median	Range	—
	3.00	2.00–2.00	1.00	0.25–1.50	0.000

*p*-value of ANOVA between PZQ ODSRTs and Aipuruike^®^.

## Data Availability

The data presented in this study are available on request from the corresponding author.

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
