# Peer review of "Preparation and In Vitro/In Vivo Evaluation of Orally Disintegrating/Modified-Release Praziquantel Tablets"

_pharmaceutics, 2021, doi:10.3390/pharmaceutics13101567_

Round 1
Reviewer 1 Report
The manuscript "Preparation and in vitro/in vivo evaluation of orally disintegrating/sustained-release praziquantel tablets" by Xuemei Wen et al. is a standard research paper presenting the development ans assessment of modified release ODTs containing praziquantel as an API. The authors describe the production process and in vitro/in vivo evaluation of proposed dosage form. That is why, in some aspects, this paper can come in handy. The paper is properly structured and the research work presented in the manuscript falls within the scope of the journal, however the authors are asked to refer to pointed below remarks.
1) During development stage it is imperative that the dosage form should be carefully designed and any potential problems involving administration should be considered. It seems that 1000 mg tablets are rather large for an ODT. It would be beneficial if the ODTs were of smaller size. E.g. IR tablets containing 34 mg is approximately 1/5 in weight, but contain nearly as much API as produced ODTs. Please provide dimensions of the obtained ODTs and rationale for using large amounts of excipients. Please comment.
2) In my opinion, the use of word 'sustained' regarding the release of an API throughout the text is misleading. The authors should consider replacing it with 'delayed release' as this term better fits the almost complete release of API within 30 min in pH = 6.8.
2) Characteristics of the process lacks the compression force used during tableting. Moreover, the ODTs should be assayed for hardness and friability.
3) To assess modified (delayed) release in vitro media change is used. E.g. 2 hours release in 0.1 N HCl and afterwards media is changed for pH 4.5 - 7.2 and dissolution study is continued. It seems that the product is well designed to retain the gastric conditions. However, if the authors would use e.g. FDA guidance, figure 4 and 5 would fit on one dissolution profile. Please comment, why the authors did not follow the guidance.
4) It is unclear how many tablets were used during in vitro study. Please refer to 'Methods', where n = 6 (line 201) and figure 5 and 6 (line 330 and line 333/334)
5) The authors stated that the most certain is that API is in amorphous state or in molecular dispersion, when processed into ODTs. What about stability of the product? Are there any signs of re-crystallization? Please comment and supply sufficient data on stability.
6) Line 386 - I do not understand why there is a question? (''How are the parameters of the HME process determined?)
Taking all of the above I recommend the manuscript to major revision.
Reviewer 2 Report
This is an interesting manuscript. I have the following comments that need to be addressed:
In the abstract you state that a range of techniques "were performed to verify successful preparation of the extrudate". These techniques do not measure preparation but they characterise the produced material - please correct this.
Line 41-42 - you use the term release rates but then provide a % - this should be the total amount dissolved at a given time point rather than a rate.
In the dissolution methodology (lines 198-201) please include the volume of dissolution media used
Rouge "247" appears in line 232 - please delete
IN your results you talk about three excipients (line 270) yet your methods only lists HPMCAS - either delete lines 270-273 or update your methods.
Explain why Aipuruike data does not appear on Figure 6 within your text
Line 363 - is it possible to include details on the oral bioavailability of PZQ from the Aipuruike formulation?
You are reporting to 4 decimal places in Table 2 - i dont think this is required - I would limit this to 2 decimal places
In your discussion on the pharmacokinetics (line 440-453) include details on what drives the efficacy of PZQ- is this Cmax or AUC driven and is there an MIC or equivalent that you can use to show why your PK profile would improve the efficacy of the product via your formulation
Round 2
Reviewer 1 Report
I would like to thank the authors for the revised version of the manuscript. I believe that corrections made by the authors would enhance the soundness and flow of the text.
I recommend to the Editor to accept the manuscript in present form.